# Comparative Study of Phenomenological Residual Strength Models for Composite Materials Subjected to Fatigue: Predictions at Constant Amplitude (CA) Loading

**DOI:** 10.3390/ma12203398

**Published:** 2019-10-17

**Authors:** Alberto D’Amore, Luigi Grassia

**Affiliations:** Department of Engineering University of Campania “Luigi Vanvitelli” Via Roma 19, 81031 Aversa (CE), Italy; luigi.grassia@unicampania.it

**Keywords:** fatigue life, residual strength, static strength, constant amplitude loading

## Abstract

The most popular methods of characterizing a composite’s fatigue properties and predicting its life are phenomenological, meaning the micro-mechanisms of composite structures under cyclic loading are not treated. In addition, in order to characterize the fatigue properties, only macro-parameters, namely strength and/or stiffness, are adopted. Residual strength models are mostly used in practice, given their strong relationship with safety and reliability. Indeed, since failure occurs when the strength degrades to the peak stress of fatigue loading, the remaining strength is used as a failure index. In this paper, based on a wide set of literature data, we summarize the capabilities of four models, namely Caprino’s, D’Amore’s, Sendekyj’s, and Kassapoglou’s models. The models are briefly described and then applied to the same data set, which is re-elaborated. The selected experimental data are recovered from a large experimental campaign carried out by the Federal Aviation Administration (FAA). Specimens of the same material were subjected to different loading in terms of peak stress, σ_max_, and stress ratio, R = σ_min_/σ_max_, ranging from pure tension (0 < R < 1) to prevalent tension (−1 < R < 0) to tension-compression (R = −1) to pure compression (1 < R < ∞). The data represent a formidable test bed to comparatively evaluate the models’ capabilities and their predictive prerogatives. The models are also tested with respect to their ability to replicate the principal responses’ feature of composite materials subjected to constant amplitude (CA) loadings. It is shown that Caprino’s and D’Amore’s models are equally capable of adequately fitting the experimental fatigue life data under given loading conditions and predicting the fatigue behavior at different loading ratios, R, with two fixed parameters. Sendekyj’s model required different parameters’ sets for each loading condition, and Kassapoglou’s model was unable to fit the majority of fatigue life data. When compared on the basis of the residual strength data, only the recently developed D’Amore’s model revealed its reliability.

## 1. Introduction

The phenomenological models for fatigue of composite materials do not provide information about their damage development. They involve macro-stress components, the cycle-by-cycle change in stiffness, and/or strength being predicted on the basis of empirical criteria. The use of phenomenological approaches constitutes a suitable solution in a structural design reality where predictive models are required for safety and reliability purposes [1,2]. One strong limit of phenomenological models is their one-dimensional character reducing the strength from a tensor to a scalar quantity. This oversimplified approach is a signature of the complexity of the phenomena underpinning fatigue where adjunctive parameters, including the tensorial nature of the strength, result in a complexity that is hardly compatible with the costs of a comprehensive experimental campaign (not to mention that the loading frequency, the test temperature, and the effect of water or other solvents are relevant parameters that would require an even larger amount of experimental efforts to secure their consistent inclusion in a model). On the other hand, the progressive damage models are based on specific failure criteria depending on the length scales and the sequence/interaction of the damage mechanisms, in principle, they may result in the development of more efficient design tools due to providing a deeper understanding of fatigue damage mechanisms. However, given the complexity of the phenomenon and the diversity of material combinations, the progressive models do not seem sufficiently robust to describe the mechanical degradation arising from damage accumulation kinetics, despite the effort spent on this task up to date [3,4]. The first residual strength degradation fatigue theory for composites was proposed by Halpin et al. and Wolff [5,6,7], who used life prediction methods for metals for guidance. However the approach was rapidly recast by a series of models that were subsequently developed taking into account that the accumulation of damage and the reduction of strength are due to the development of a multiplicity of degradation mechanisms operating at different length scales and not to a single crack propagation. The majority of these models, at least those that are mostly accredited, were reviewed comprehensively by Philippidis and Passipoularidis [8,9] to check their reliability for a series of carbon/epoxy and glass/epoxy laminates subjected to fatigue. They concluded that some models were only occasionally able to predict the residual strength under particular loading conditions of different laminates, an outcome that prevented the widespread use of such models. In all cases, the statistical predictions at different stress levels and fractions of fatigue life were highly unreliable. They concluded that simple models, namely those with a limited number of parameters and as such requiring limited experimental efforts, are preferable compared to complicated multiparametric ones where vast experimental campaigns are often needed to optimize the models’ parameters. Furthermore, the majority of the models they reviewed did not account for the loading ratio R = σ_min_/σ_max_, or that when R is considered, several additional parameters are required. This in most cases prevented the predictive use of such models out of the experimentally visited loading conditions and, more importantly, the adoption of such models under variable amplitude (VA) loadings. However, in general the basic equations of residual strength models correlate the strength with the number of cycles, n, the maximum stress, σ_max_, the loading ratio, R = σ_min_/σ_max_, and the static strength, $\sigma_{0}$, resulting in the following implicit expression:(1)$$\frac{d\sigma_{r}}{dn}=f^{\prime }\left(n,\;\sigma_{max},\;R,\;\sigma_{0}\right).$$

Through integration of Equation (1), the residual strength explicit equation is recovered and the equation for the fatigue life, namely the S-N behavior, can be derived from as a limiting case, namely when the residual strength degrades to the maximum applied stress $\sigma_{r}=\sigma_{max}\;\mathrm{and}\;n=N$, with *N* being the number of cycles until failure occurs. Accordingly, by integrating Equation (1), the equation for fatigue life can be written as follows: (2)$$\sigma_{e}=\sigma_{0}=f\left(N,\;\sigma_{max},\;R\right)$$

Generally, the explicit function f describes the “nominal” or “mean” fatigue life behavior, with the statistical nature of the fatigue response being almost completely affected by the statistical distribution of static strength. Equation (2) also defines the “equivalent strength” models where $\sigma_{e}$ is the model-generated nominal strength. In fact, one criterion to evaluate the robustness of a model [10,11,12,13,14,15,16,17,18,19] is to verify that the equivalent strength, $\sigma_{e}$, replicates the statistics of the static strength, $\sigma_{0}$, that follows a Weibull probability distribution function:(3)$$P_{\sigma }=P\left(X\leq \sigma \right)=1-exp\left[-{\left(\frac{\sigma }{\gamma }\right)}^{\delta }\right]$$
where *γ* and *δ* are the scale and shape parameters of the distribution function, respectively. 

In that case, Equation (2) self-consistently fulfills the requirements based on a strength-life equal rank assumption (SLERA), a concept stating that not only do stronger samples have a longer life expectation but also that samples of a given rank in the distribution function of static strength should have the same rank in the distribution function of fatigue life, under given loading conditions. The concept, first assumed by Han and Kim [20] and then recast by Chou and Croman [21], remains hardly demonstrable (in fact, the original static strength of samples failed under fatigue remains unknown) and represents the substantial physics behind any reliable model. 

Under the above conditions, the statistics of fatigue life distribution function can be recovered by using Equations (2) and (3) as follows:(4)$$F_{N}\left(n\right)=P_{\sigma }=P(X\leq \sigma ,\;N<N^{*})=1-exp\left[-{\left(\frac{f\left(N,\;\sigma_{max},R\right)}{\gamma }\right)}^{\delta }\right]$$
where $F_{N}\left(n\right)$ is the probability function used to find a case where *N* < *N**. The explicit function f and the shape and scale parameters, *γ* and *δ*, are the basis for calculations. The Weibull distribution function of number of cycles to failure, under given loading conditions, can be written as follows:(4’)$$F_{N}\left(n\right)=P_{\sigma }=P\left(N<N^{*}\right)=1-exp\left[-{\left(\frac{N^{*}}{\eta }\right)}^{\mu }\right]$$
where η and μ are the scale and the shape factors. Equations (4) and (4’) show that the scale and the shape factors of static strength and fatigue life distribution functions are strongly correlated for a given material. Equation (4) states that the distribution function of the fatigue life, namely Equation (4’), is “scaled” by the fatigue life function, f. However, it was shown that while the shape factor of the static strength distribution may attain values between 15 and 60 [22], the shape factor for fatigue life varies between 0.8 and 2.

In general, the fatigue life curves are expressed by deterministic equations with at least two fitting parameters. Residual strength models also comply with experimental evidence showing that a fatigue limit seems unlikely in composite materials or irrelevant in the range of cycles encountered in practical applications, meaning the slope of S–N curves should remain always negative. Starting from the rate Equation (1), the above general approach was followed progressively by several authors. Many refinements and models’ predictions have appeared so far [9]. In a comprehensive review by Philippidis and Passipoularidis [9], it was reported that most of phenomenological models were occasionally suitable for safety and reliability purposes. Based on experimental data available in literature, in the following section we compare four phenomenological models that were not considered before in relation to each other, namely Sendekyj’s [11], Kassapogoulos’ [23], Caprino’s [14], and D’Amore’s [16] models. The selection of such models was driven by the following different considerations: (1) Sendekyj’s model is widely used for safety and reliability purposes in industrial environments, especially in aerospace (2) Kassapogoluos’ model claims to predict the fatigue life of any materials starting from the determination of scale and shape parameters of the static strength distribution function, (3) Caprino’s model incorporates the loading ratio, R, and was applied with success to predict the fatigue life of a number of different composites categories [24,25,26,27,28,29,30], (4) D’Amore’s model is an extension of Caprino’s model. In its modified version, D’Amore’s model was proven to be able to predict the cycle-by-cycle fatigue damage accumulation and the residual strength of composites subjected to variable amplitude loading. The four models are described in the following section and their capability is illustrated on the basis of the same data set. 

## 2. Residual Strength Models 

### 2.1. Sendeckyj’s Model

The basic Sendeckyj’s model [11] is presented in the deterministic equation given by
(5)$$\sigma_{e}=\sigma_{max}{\left[{\left(\frac{\sigma_{r}}{\sigma_{max}}\right)}^{\frac{1}{S}}+\left(n-1\right)C\right]}^{S}$$
where *σ_e_* is the equivalent static strength, *σ_max_* is the maximum applied cyclic stress, *σ_r_* is the residual strength, *n* is the number of fatigue cycles, and *S* and *C* are fitting parameters. From Equation (5), the residual strength can be calculated as follows:(5’)$$\sigma_{max}{\left[{\left(\frac{\sigma_{e}}{\sigma_{max}}\right)}^{\frac{1}{S}}+\left(n-1\right)C\right]}^{S}=\sigma_{r}.$$

Sendeckyj’s equivalent static-strength model assumes that the strongest specimen has the longest fatigue life or the highest residual strength at run-out. From Equation (5) assuming that failure occurs when the residual strength degrades to the maximum cyclic stress, the following deterministic expression for fatigue life can be obtained: (5”)$$\sigma_{max}{\left(1-C+Cn_{f}\right)}^{S}=\sigma_{u}$$
where *σ_u_* is the static strength and $n_{f}$ is the number of cycles before failure. Using Equation (5”) requires the optimization of different parameters’ sets for each loading ratio, R. Once calculated, the model’s parameters allow us to convert the fatigue life and residual strength data into an equivalent static strength data set, namely $\sigma_{e},$ to be fitted into a Weibull distribution function [6]. In order to account for a different loading ratio, *R*, several versions of Equation (6) were developed and summarized in Table 1. 

### 2.2. Kassapogoulos’ Model

Starting from basic statistics and the strong assumption that the probability of failure is constant with the number of cycles, Kassapogolou [23] derived a very simple and attractive model for fatigue life. Based on the knowledge of the Weibull’s shape and scale parameters of the static strength, the stress σ that will lead to failure at N cycles was described by the following Equation:(6)$$\sigma =\frac{\gamma }{N^{\frac{1}{\delta }}}$$
with *δ* and *γ* as the shape and the scale parameters of the two-parameter Weibull distribution describing the static strength, respectively. Equation (6) is essentially an expression for the S-N curve of the structure and was derived for the cases when 0 ≤ R <1 or R > 1. When R < 0, the S–N expression takes the following form:(7)$$N=\frac{1}{{\left(\frac{\sigma_{max}}{\gamma_{T}}\right)}^{\delta_{T}}+{\left(\frac{\sigma_{min}}{\gamma_{C}}\right)}^{\delta_{C}}}.$$
with $\gamma_{T}$, $\delta_{T}$ and $\gamma_{C}$, $\delta_{C}$ as the Weibull parameters of the static strength in tension and compression, respectively. For the particular case of $R=\frac{\sigma_{min}}{\sigma_{max}}=-1:$(8)$$N=\frac{1}{{\left(\frac{\sigma }{\gamma_{T}}\right)}^{\delta_{T}}+{\left(\frac{\sigma }{\gamma_{C}}\right)}^{\delta_{C}}}.$$

Thus, Equation (7) requires iterative methods to be solved. The substantial claim of the model is that it only requires the knowledge of static strength data to fully predict the fatigue behavior (a case that appears unrealistic at first glance).

### 2.3. Caprino’s Model

In this section we summarize the principal features of the wear-out model already proposed in reference [14]. The model is expressed by the following equations for residual strength and fatigue life, respectively: (9)$$\sigma_{n}=\sigma_{o}-\alpha \sigma_{max}\left(1-R\right)\left(n^{\beta }-1\right)$$
(9’)$$\sigma_{0N}=\sigma_{0}=\sigma_{\max}\left[\alpha \left(1-R\right)(N^{\beta }-1)+1\right]$$
where *n* is the current cycle, $\sigma_{\max}\left(1-R\right)=\Delta \sigma =(\sigma_{\max}-\sigma_{\min})$ is the amplitude of cyclic loading, $R=\frac{\sigma_{\min}}{\sigma_{\max}}$ is the loading ratio, N is the number of cycles to failure, and *α* and *β* are the model parameters. $\sigma_{0N}$ represents the “virgin strength” of samples fatigued until failure and coincides with the experimentally determined static strength statistics, σ_0_, represented by a two-parameter Weibull distribution as follows:(10)$$F_{\sigma_{0}}\left(x\right)=P\left(\sigma_{0}\leq x\right)=1-\exp\left[-{\left(x/\gamma \right)}^{\delta }\right]$$

By means of Equation (9’), the virgin strength, $\sigma_{0N}$, of samples subjected to fatigue can be recovered, under fixed $\sigma_{max}$ and R. Therefore, from a series of fatigue life data, the statistics of static strength can be obtained. Accordingly, Equations (9’) and (10) can be triggered to determine the Weibull statistics of the number of cycles to failure as follows:(11)$$F_{N}\left(n\right)=P\left(N\leq n\right)=1-\exp\{-{\left[\frac{\sigma_{\max}\;\left[1+\alpha \left(n^{\beta }-1\right)\left(1-R\right)\right]}{\gamma }\right]}^{\delta }\}$$
where $F_{N}\left(n\right)$ is the cumulative distribution function. We recall that model parameters α and β, as well as Weibull parameters γ and δ, remain fixed along the calculations. Moreover, when n = 1 the Weibull statistics for the static strength is recovered, namely Equation (10). In summary, the model accounts for the loading ratio, R, a prerogative that implies a predictive capability of fatigue life from experimental conditions needed to fix the model’s parameters, α and β. On the other side, the Caprino’s model revealed its unreliability in predicting the strength degradation during fatigue [13,14].

### 2.4. D’Amore’s Model

D’Amore’s model differs conceptually from Caprino’s model as it is assumed that any sample preserves the same rank within the statistics of static strength, fatigue life, and residual strength [15,16,17]. In other words, while Caprino’s model fulfils the strength-life equal rank assumption (SLERA) [6], the extension of the concept to residual strength statistics can be recovered in the framework of D’Amore’s approach as follows:(12)$$P_{REL,\sigma_{max}}\left(X\geq \sigma_{in}\right)=1-P\left(N\leq n\right)=exp\{-{\left[\frac{\sigma_{max}\;\left[1+\alpha \left(n^{\beta }-1\right)\left(1-R\right)\right]}{\gamma }\right]}^{\delta }\}$$
where $P_{REL,\sigma_{max}}\left(X\geq \sigma_{n}\right)$ is the reliability function, which accounts for the probability of finding a specimen with strength $\left(X\geq \sigma_{in}\right)$, under fixed loading conditions, namely R and $\sigma_{\max}$. Thus Equation (12) can readily converted to the description of the strength evolution of a single specimen with the current number of cycles, *n*. The approach is straightforward: during fatigue the generic specimen of a given rank within the static strength distribution function degrades its strength from the original strength, $\sigma_{i_{0N}}$, at n=1, towards the maximum applied stress, namely, when n = N_i_, $\sigma_{i_{n}}\left(N_{i}\right)=\sigma_{max}$. Therefore, the strength degradation function assumes the following form:(13)$$\frac{\sigma_{i_{n}}-\sigma_{max}}{\sigma_{i_{0N}}-\sigma_{max}}=exp\{-{\left[\frac{\sigma_{max}\;\left[1+\alpha \left(n^{\beta }-1\right)\left(1-R\right)\right]}{\gamma_{i}\left(\sigma_{i_{0N}}\right)}\right]}^{\delta }\}$$
where
(14)$$\gamma_{i}\left(\sigma_{i_{0N}}\right)=\frac{\sigma_{i_{0N}}}{\sigma (\gamma )}\gamma$$
is the scaling factor for the *i*-th sample with a “virgin” strength of $\sigma_{i_{0N}}$. For convenience, Equation (13) expresses the need to consider that weaker and stronger samples cannot exhibit the same characteristic decay strength, which is $\gamma_{i}\left(\sigma_{i_{0N}}\right)$. Furthermore, it is assumed that the static strength distribution can be realistically confined between the extreme ranks F = 0.95 and F = 0.05, namely the arbitrarily defined upper and lower tails of the distribution function (however any other limits can be defined). Thus, during fatigue we assume that the strength of stronger sample with F = 0.95 should degrade down to the maximum applied stress, $\sigma_{max}$, with a characteristic decay strength, γ namely the Weibull shape factor of the static strength distribution function. 

Indeed, Equation (13) states that ratio of the samples with different ranks equals the ratio of their characteristic decay strength and, while this assumption works very well, a refinement of it is under study. From above, it can be appreciated that stronger sample may degrades towards a strength, $\sigma_{j}\left(n\right)=\sigma_{i_{0N}}$, with $\sigma_{i_{0N}}$ being the virgin strength of a weaker sample. This is a sign that the strength itself is not a sensitive measure of the state of damage, as the accumulation of damage is different with samples of a different static strength. The degradation curves departing from strengths of rank F = 0.95 and 0.05 actually represent the upper and lower bounds of strength domain when the remaining strength is measured under the same loading condition at a given number of cycles. Finally, from Equation (13), the formal expression for the strength degradation kinetics is: (15)$$\frac{{d\sigma }_{i_{n}}}{\mathrm{dn}}=-\left(\sigma_{i_{0N}}-\sigma_{\max}\right)\frac{\exp\left[-{\left(\frac{A}{\gamma_{i}\left(\sigma_{i_{0N}}\right)}\right)}^{\delta }\right]n^{\beta -1}\left(1-R\right){\alpha \beta \delta \sigma }_{\max}{\left(\frac{A}{\gamma_{i}\left(\sigma_{i_{0N}}\right)}\right)}^{\delta -1}}{\gamma_{i}\left(\sigma_{i_{0N}}\right)}$$
where
(16)$$A=\left[1+\left(n^{\beta }-1\right)\left(1-R\right)\alpha \right]\sigma_{max}$$
Meaning the equation’s parameters are already defined.

From Equation (13) it can be seen that when *n*
≅
*N*, namely $\sigma_{\mathrm{in}}\approx \sigma_{\max}$, Equation (9’) of the Caprino’s model is obtained as follows:(17)$${\left[-\ln\frac{\sigma_{n}-\sigma_{\max}}{\sigma_{\mathrm{ON}}-\sigma_{\max}}\right]}^{\frac{1}{\delta }}=\frac{\sigma_{\max}\;\left[1+\alpha \left(N^{\beta }-1\right)\left(1-R\right)\right]}{\gamma }≅1$$

Thus, Equation (17) for fatigue life is a limiting case of Equation (13), namely the residual strength equation. In a form suitable to fitting the experimental S-N data, Equation (17) can be written as follows:(18)$$\sigma_{\max}=\sigma_{0}{\left[\alpha \left(N^{\beta }-1\right)\left(1-R\right)+1\right]}^{-1}$$
where N is the number of cycles to failure at a given $\sigma_{\max}$ and $\sigma_{0}$ = γ. 

Equation (18) is identical to Equation (9’), thus the substantial difference between Caprino’s and D’Amore’s model lies in Equations (9) and (13), respectively. It is worth noting that the parameters of Caprino’s and D’Amore’s models are also identical, a case that allows recovering the large amount of fatigue life predictions previously obtained with no modifications [14,15,16,17,18,19,22,24,25,26,27,28,29,30]. 

## 3. Experimental Data 

The experimental data are recovered from a technical report from FAA. [31]. We selected a series of fatigue and residual strength data performed under various loading condition on AS4 carbon/epoxy 3k/E7K8 Plain Weave Fabric with [45/−45/90/45/−45/45/−45/0/45/−45]_S_ layup. The experimental campaign consisted of fatigue tests performed at three levels of stresses, six static and three residual strength tests, following international standards (ASTM D5766). In this paper we report the predictions of Caprino’s, D’Amore’s, Sendeckyj’s, and Kassapogoulos’ models towards the data obtained on “open hole” (OH) specimens subjected to prevailing tension or compression tests at different loading ratios of R. Deeper details on materials and specimen geometry can be found in reference [31]. The contract notation 10/80/10 will be used in what follows representing a laminate with 10% laminae orientated at 0 and 90 degrees, respectively, with the resting 80% being oriented at ±45 degrees.

The capability of the different models is illustrated first in Figure 1 towards fatigue life data where given the loading conditions, namely R = 0 and R = −0.2, tension stresses are prevailing. The black lines refer to the equivalent models by Caprino and D’Amore. The broken curves represent the best fitting curves to the data of the Sendeckyj’s model from one side, and the predictions based on the Kassapogoulos’ model (let us recall that the Kassapogoulos’ model only requires the static strength data for predictions), as indicated. The predictions based on Caprino’s and D’Amore’s models are obtained with fixed parameters while Sendekyj’s and Kassapogolous’s models require a different set of parameters for each loading condition. The same general considerations can be done for Figure 2, reporting the fatigue data obtained under prevailing compressive stress, namely when R = −1 and R = 5. To this end it must be recognized that despite the symmetry of the loading condition at R = −1, namely the same absolute peak stresses are applied in tension and compression, the final collapse occurs in compression. This is not surprising given that the static strength in compression is lower than that in tension. This experimental evidence can be appreciated when comparing the static strength data (namely, those data at n = 1) in Figure 1 and Figure 2.

From the above data, Sendekyj’s, Caprino’s, and D’Amore’s models seem equivalent (also because at longer cycles the model are almost superimposed). This is not true substantially because the analytic Sendekjy’s model required different sets of parameters for each loading condition, namely R, while the parameters of Caprino’s and D’Amore’s models remain fixed.

Concerning the Kassapogolous’ model, it is apparent that the fortuitous fitting of data at R = 5 does not guarantee an acceptable reliability. To illustrate the powerfulness of the Caprino’s and D’Amore’s models, the complete set of data is reported in Figure 3. The best fit to the data obtained at R = 0 and R = −1 allowed us to calculate the model’s parameters α and β that remained fixed to fully predict the fatigue behavior under different loading condition, as indicated. It is worth mentioning that the models parameters differ when prevailing compression or tension loadings are in play. The reason for this is that the properties degradation in compression and tension may follow different paths given the adjunctive damage mechanisms occurring during the compression loadings that may dictate a different hierarchy of damage development. Thus, Equation (9’) is used two times pooling the static strength and fatigue data at R = 0 and R = −1 separately, and using means of best fit procedures allows recovering the model’s parameters in tension, namely α_t_ and β_t_, and the prevailing compression (R = −1), namely α_c_ and β_c_. The two parameters’ set allow us to fully predict the fatigue behavior at R = −0.2 and R = 5, where the tension and compression are prevailing loading modes, respectively. 

Completely different considerations arise when predictions of the strength after a given number of cycles are in order. For instance, Figure 4 and Figure 5 report the fatigue life and the residual strength data for open hole (OH) specimens tested at different loading ratios, namely R = −1 and R = −0.2, where prevalent compression and tension loadings are in play, respectively (at R = −1, despite the symmetry of loading, the rupture of samples occurs in compression since the compression strength is lower than in tension given the stacking sequence of the OH specimens under study). 

In both figures the equivalent D’Amore’s and Caprino’s models for fatigue life (continuous curves) are used to fix the models’ parameters. Then, based on Equations (13) and (14), the predictions of D’Amore’s model are reported as dotted red curves. The data remain confined within the domain bounded by upper and lower residual strength curves, namely the degradation curves departing from static strengths of rank F = 0.95 and 0.05, respectively. The unreliability of the Caprino’s model was already ascertained before [28,29,30]. However, for completeness, the predictions based on Equation (9) are also reported in Figure 4 and Figure 5 as black dotted curves. Concerning the Sendekyj’s model, based on Equation (5’’) the predictions are represented as dotted blue curves. However, we must emphasize that this model was rarely used for residual strength predictions. The reason is that, differently from Caprino’s and D’Amore’s models, the Sendekyj’s model does not inherently fulfill the failure condition, namely σ_r_ = σ_max_ at n = N, as can be readily seen in Figure 4 and Figure 5. On the other side, the Kassapoglous’s model is unreliable even for fatigue life predictions. 

The capabilities of the D’Amore’s model under constant amplitude (CA) loadings were already reported [15,16,17,18,19,22] for very different composites categories. With two fixed parameters, the model is capable of treating the statistical nature of fatigue response and recovers the static strength from fatigue life data in a way where it consistently obeys the strength-life equal-rank assumption (SLERA). This means it allows for predicting the fatigue life under different loading conditions incorporating the loading ratio, R, and, ultimately it is able to predict the residual strength with no parameter adjustments [18]. Moreover, the model was recently applied with success to the case of very complex spectrum loadings [19]. For clarity, the different model’s capabilities discussed above are summarized in Table 2. In this Table, we highlight that Caprino’s and D’Amore’s models require the optimization of only two parameters to predict the fatigue life of composite materials under different loading scenarios and, while Caprino’s, Kassapoglous’s and Sendekyj models are unreliable in predicting the residual strength, D’Amore’s approach allows predicting the evolving strength with fixed parameters set with no adjustments. In particular, D’Amore’s model captures the “sudden drop” behavior of strength evolution during fatigue. Thus, fulfilling the principal features of fatigue responses with a limited parameters set allows an easy use of D’Amore’s model for the optimization of test campaign and the development of generalized softwares that can be adopted for predictions under variable amplitude (VA) loadings on sound basis. Moreover, the analytical differences between Caprino’s and D’Amore’s models are shown in Table 3. As already mentioned, the two models coincide when fatigue life predictions are a concern, whereas the residual strength degradation equations greatly differ and so do the rate equations. 

## 4. Conclusions

A comparative study was performed to check the reliability of four phenomenological models in predicting the fatigue life and residual strength of composite materials. A coherent set of experimental data taken from literature was used for comparison. It is shown that Caprino’s, Sendekyj’s, and D’Amore’s phenomenological models for fatigue life are almost equivalently reliable. However, the Sendekyj’s model required different sets of model parameters when a different loading ratio, R, was in play. This prevented any predictive capability of Sendeckyj’s model when different loading scenarios were in play. Instead Caprino’s and D’Amore’s fatigue life models incorporated the loading ratio, R, and simply required a set of experimental fatigue data to fix the two model’s parameters, namely α and β. It is shown that the models fully predict the fatigue responses under different loading conditions accounting for the stress ratio, R, without recurring to parameters adjustments. Kassapoglou’s model revealed its inadequacy by only occasionally fitting the data. Furthermore, it was evident that both the Caprino’s and Sendekyj’s models were unable to predict residual strength data. Instead, the recently developed D’Amore’s model showed its full reliability. The potential of D’Amore’s model is readily ascertained, recalling that the fatigue life model is, in fact, a particular case of the residual strength model. Furthermore, D’Amore’s model can replicate the principal features of composites subjected constant amplitude (CA) loadings, making it a strong candidate to be evaluated when spectrum loadings are a concern.

## Figures and Tables

**Figure 1 materials-12-03398-f001:**
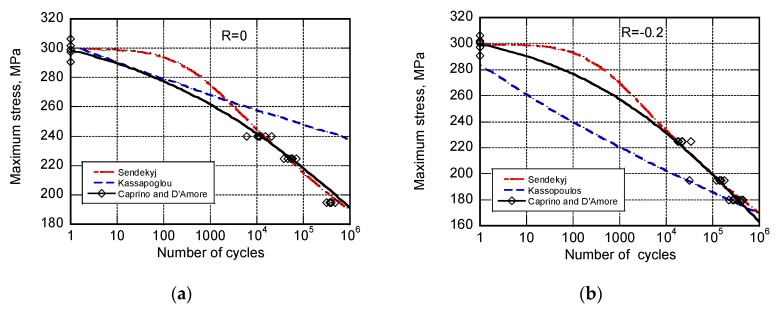
The experimental fatigue life data for AS4/E7K8 PW—10/80/10, OH, at R = 0 (**a**) and R = −0.2 (**b**) and the predictions of different models as indicated in the inset.

**Figure 2 materials-12-03398-f002:**
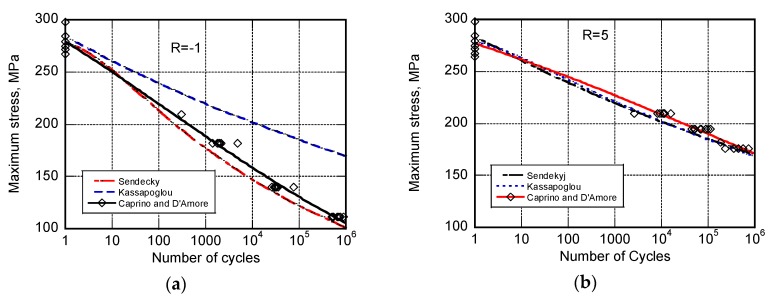
The experimental fatigue life data for AS4/E7K8 PW—10/80/10, OH, at R = −1 (**a**) and R = −5 (**b**) and the predictions of different models as indicated in the inset.

**Figure 3 materials-12-03398-f003:**
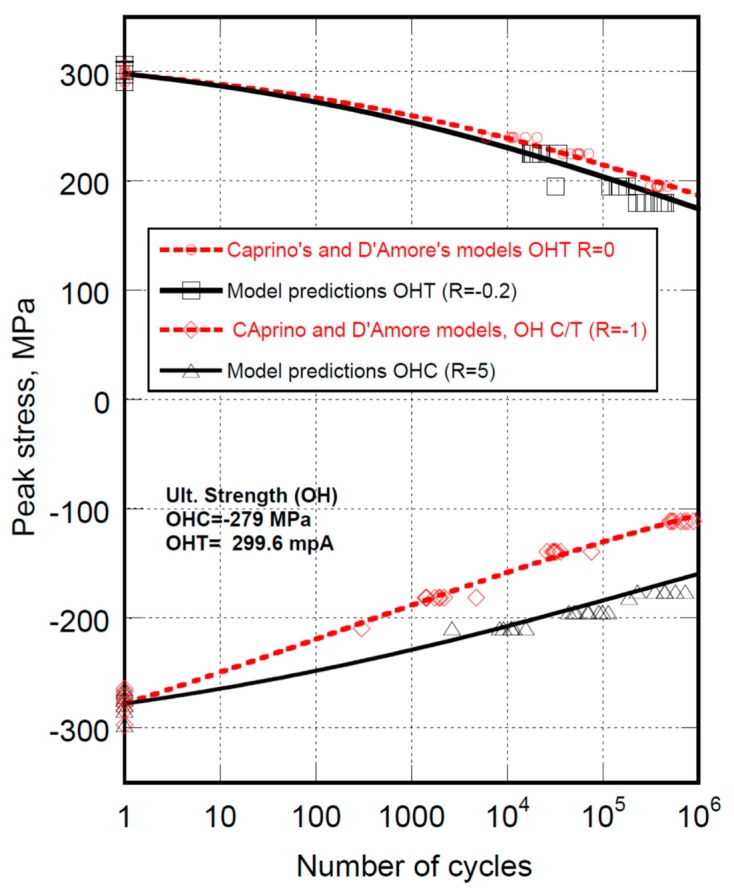
The complete set experimental fatigue life data for AS4/E7K8 PW—10/80/10, OH, at R = 0, R = −0.2, R = −1 and R = −5 and the predictions of equivalent D’Amore’s and Caprino’s models. The data are modeled with a single set of models parameters.

**Figure 4 materials-12-03398-f004:**
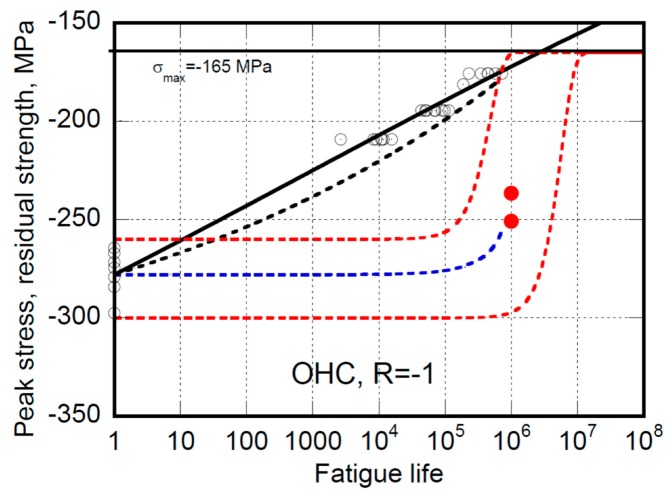
The experimental fatigue life (empty circles) and residual strength data (filled circles) for open hole AS4/E7K8 PW—10/80/10 under prevalent compression loadings (OHC) at R = −1. The continuous curve is the best fit to the fatigue life data based on Caprino’s and D’Amore’s equivalent models. The black broken curve is the predictions of residual strength from Caprino’s model, namely Equation (9). The red broken curves are the predictions of upper and lower bound of residual strength from the D’Amore’s model, namely Equation (12). The blue dotted line represents the Sendekyj’s model’s residual strength predictions, namely Equation (5’’).

**Figure 5 materials-12-03398-f005:**
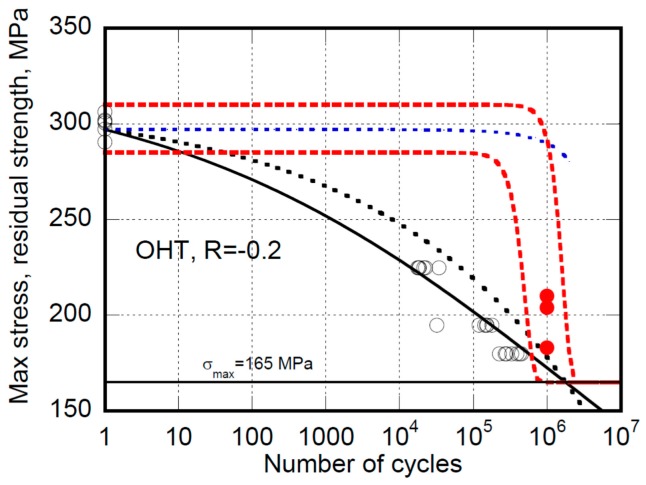
The experimental fatigue life (circles) and residual strength (filled circles) data for open hole AS4/E7K8 PW—10/80/10, under prevalent tension (OHT) at R = −0.2. The continuous curve is the best fit to the fatigue life data based on Caprino’s and D’Amore’s equivalent models. The broken black curve is the predictions of residual strength from Caprino’s model, namely Equation (9). The broken red curves are the predictions of upper and lower bound of residual strength from the D’Amore’s model, namely Equation (13). The blue dotted line represents the Sendekyj’s model’s residual strength predictions, namely Equation (5’’).

**Table 1 materials-12-03398-t001:** Sendeckyj’s additional models for fatigue life predictions.

Parameters		
-	s	f
W1	S_0_	1
W2	S_0_	C
W3	S_0_	C(1 − R)^G^
W3A	S_0_(1 − R)^G^	C(1 − R)^G^
W4	S_0_ + D(1 − R)^G^	C(1 − R)^G^
W4A	S_0_(1 − R)^G^	C(1 − R)^G^

**Table 2 materials-12-03398-t002:** Comparative capabilities of the relevant models.

-	Number of Parameters	Recovery of The Static Strength Distribution from Fatigue Data (The Concept of Equivalent Static Strength)	Predictions of Fatigue Life under Different Loading Conditions	Prediction of “Sudden Drop” of Strength with Fixed Parameters Optimized on The Basis of Fatigue Life and Static Strength Data	Easy Use for Reliability Analysis (Based on Few, Fixed Parameters)	Easy Use for The Optimization Of Test Campaign. The Development of Generalized Softwares and Predictions under Variable Amplitude Loadings
D’Amore (2015)	2	Excellent	Excellent	Excellent (no adjustments allowed). The parameters are fixed once and for all.	Excellent	Excellent
Caprino (1996)	2	Excellent	Excellent	poor	poor	poor
Sendekyj (1980)	>2 each loading condition	Excellent	Poor: Only fit to the data (no predictions). Different set of parameters each loading condition	poor	poor	poor
Kassapouglos (2011)	No parameters needed. Only the two parameters of the Weibull distribution function are claimed to be sufficient to predict the fatigue life (No fatigue life data are required)	Not shown	Prediction are un-conservative. Fatigue curves are shifted some 2 or more decades of cycles in respect to the experimental data	Not shown	Not shown	Not shown

**Table 3 materials-12-03398-t003:** Comparison of Caprino’s and D’Amore’s models.

-	Equation for The Rate of Strength Degradation	Residual Strength Evolution Equation	Fatigue Life Equation
D’Amore (2015)	$\frac{d\sigma_{i_{n}}}{dn}=-\left(\sigma_{i_{0N}}-\sigma_{max}\right)\frac{exp\left[-{\left(\frac{A}{\gamma_{i}\left(\sigma_{i_{0N}}\right)}\right)}^{\delta }\right]n^{\beta -1}\left(1-R\right)\alpha \beta \delta \sigma_{max}{\left(\frac{A}{\gamma_{i}\left(\sigma_{i_{0N}}\right)}\right)}^{\delta -1}}{\gamma_{i}\left(\sigma_{i_{0N}}\right)}$	$\frac{\sigma_{i_{n}}-\sigma_{max}}{\sigma_{i_{0N}}-\sigma_{max}}=exp\{-{\left[\frac{\sigma_{max}\;\left[1+\alpha \left(n^{\beta }-1\right)\left(1-R\right)\right]}{\gamma_{i}\left(\sigma_{i_{0N}}\right)}\right]}^{\delta }\}$	$\sigma_{max}\;\left[1+\alpha \left(N^{\beta }-1\right)\left(1-R\right)\right]≅\sigma_{oN}$
Caprino (1996)	$\frac{d\sigma_{n}}{dn}=-a_{0}\Delta \sigma n^{-b}$	$\sigma_{n}=\sigma_{0}-\sigma_{\max}\alpha \left(1-R\right)\left(n^{\beta }-1\right)$	$\sigma_{max}\;\left[1+\alpha \left(N^{\beta }-1\right)\left(1-R\right)\right]≅\sigma_{oN}$

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
