# Peer review of "Comparative Study of Phenomenological Residual Strength Models for Composite Materials Subjected to Fatigue: Predictions at Constant Amplitude (CA) Loading"

_materials, 2019, doi:10.3390/ma12203398_

Round 1

Reviewer 1 Report

Authors compared the four different phenomenological model and compared the fit. Authors have explained all the models in detail that shows that two models predict the fatigue behavior accurately. 

I have recently reviewed a similar kind of study from the same group.

This manuscript will be helpful for the researchers working in the modeling of fatigue failure. I will recommend accepting this manuscript with just minor changes. The figures are not looking that clear, redraw the figures.

Author Response

ANSWERS

Authors compared the four different phenomenological model and compared the fit. Authors have explained all the models in detail that shows that two models predict the fatigue behavior accurately. 

ANSWER: The fatigue life behavior was accurately predicted by Caprino's, Sendekyj's and D'Amore's  models. The residual strength data were predicted only by the D'Amore's model.

    2. I have recently reviewed a similar kind of study from the same group.

ANSWER: Yes, in some way this is true concerning the D'Amore's model but a comparative study among the different models  capabilities was never done.

  3.  This manuscript will be helpful for the researchers working in the modeling of fatigue failure. I will recommend accepting this manuscript with just minor changes. The figures are not looking that clear, redraw the figures

ANSWER: Some amendments are reported in the revised manuscripts and some figures were clarified with amended captions.

Reviewer 2 Report

Dear Authors,

The aim of this study was to compare four residual strength models for composites subjected to fatigue. The prediction of fatigue life is important to develop new composites. While the topic is fitting to the journal scope, some minor concerns were raised. Revise the manuscript by following comments.

Minor points

Page 2, Line 64

“carbo/epoxy” may be the typo. It should be modified to “carbon/epoxy”.

Page 2, Line 66

“use use” is the typo. It should be modified to “use”.

Page 3, Line 130

“a number of a number of” is the typo. It should be modified to “a number of”.

Page 5, Equation 23

“exp” should be written in Italic font? Make sure all other related points.

Figure 4 and Figure 5

Graph legends must be prepared.

Author Response

The typos were corrected and a a list of figure captions was prepared

Reviewer 3 Report

In this work, the authors introduced the capabilities of four residual strength models, and made a comparative study by replicating the principal responses’ feature of composite materials subjected to constant amplitude (CA) loadings. A coherent set of experimental data taken from literature was used for comparison. The topic of current research should be of interest for the field, however, the following comments should be addressed for further consideration.

The novelty of this work needs to be strengthened, the motivation of comparative study on those four models are not clear; In Section 3, more details in experiments should be given, especially for test conditions. The legends in Figs. 4 and 5 should be added; The main conclusion points from Tables 2 and 3 should be clarified and elaborated in the text; The sub-title of “3.9” and “3.10” should be “2.3” and “2.4”.

Author Response

In this work, the authors introduced the capabilities of four residual strength models, and made a comparative study by replicating the principal responses’ feature of composite materials subjected to constant amplitude (CA) loadings. A coherent set of experimental data taken from literature was used for comparison. The topic of current research should be of interest for the field, however, the following comments should be addressed for further consideration.

The novelty of this work needs to be strengthened, the motivation of comparative study on those four models are not clear; In Section 3, more details in experiments should be given, especially for test conditions. The legends in Figs. 4 and 5 should be added; The main conclusion points from Tables 2 and 3 should be clarified and elaborated in the text; The sub-title of “3.9” and “3.10” should be “2.3” and “2.4”. 

ANSWER: The novelty was highlighted in the text. The comparative study is motivated by the fact that there is only one model (the D'Amore's model) that is capable to predict simultaneously the fatigue life and the residuals strength of composite materials .

The sequence of paragraph was adjusted and the figures captions completed.  

Round 2

Reviewer 3 Report

All comments have been addressed in this revision, which can be accepted for publication as it is.